# Role of *Mn-LIPA* in Sex Hormone Regulation and Gonadal Development in the Oriental River Prawn, *Macrobrachium nipponense*

**DOI:** 10.3390/ijms25031399

**Published:** 2024-01-23

**Authors:** Pengfei Cai, Wenyi Zhang, Sufei Jiang, Yiwei Xiong, Hui Qiao, Huwei Yuan, Zijian Gao, Yongkang Zhou, Shubo Jin, Hongtuo Fu

**Affiliations:** 1Wuxi Fisheries College, Nanjing Agricultural University, Wuxi 214081, China; ckgg5436@126.com (P.C.); yuan08102021@126.com (H.Y.); gaozijiangenomics@163.com (Z.G.); 18555601471@163.com (Y.Z.); 2Key Laboratory of Freshwater Fisheries and Germplasm Resources Utilization, Ministry of Agriculture and Rural Affairs, Freshwater Fisheries Research Center, Chinese Academy of Fishery Sciences, Wuxi 214081, China; zhangwy@ffrc.cn (W.Z.); jiangsf@ffrc.cn (S.J.); xiongyw@ffrc.cn (Y.X.); qiaoh@ffrc.cn (H.Q.)

**Keywords:** *Macrobrachium nipponense*, *Mn-LIPA*, sex hormone, gonadal development, reproduction, RNAi

## Abstract

This study investigates the role of lysosomal acid lipase *(LIPA)* in sex hormone regulation and gonadal development in *Macrobrachium nipponense*. The full-length *Mn-LIPA* cDNA was cloned, and its expression patterns were analyzed using quantitative real-time PCR (qPCR) in various tissues and developmental stages. Higher expression levels were observed in the hepatopancreas, cerebral ganglion, and testes, indicating the potential involvement of *Mn-LIPA* in sex differentiation and gonadal development. In situ hybridization experiments revealed strong *Mn-LIPA* signaling in the spermatheca and hepatopancreas, suggesting their potential role in steroid synthesis (such as cholesterol, fatty acids, cholesteryl ester, and triglycerides) and sperm maturation. Increased expression levels of male-specific genes, such as insulin-like androgenic gland hormone *(IAG)*, sperm gelatinase *(SG)*, and mab-3-related transcription factor *(Dmrt11E)*, were observed after *dsMn-LIPA* (double-stranded *LIPA*) injection, and significant inhibition of sperm development and maturation was observed histologically. Additionally, the relationship between *Mn-LIPA* and sex-related genes (*IAG*, *SG*, and *Dmrt11E*) and hormones (17β-estradiol and 17α-methyltestosterone) was explored by administering sex hormones to male prawns, indicating that *Mn-LIPA* does not directly control the production of sex hormones but rather utilizes the property of hydrolyzing triglycerides and cholesterol to provide energy while influencing the synthesis and secretion of self-sex hormones. These findings provide valuable insights into the function of *Mn-LIPA* in *M. nipponense* and its potential implications for understanding sex differentiation and gonadal development in crustaceans. It provides an important theoretical basis for the realization of a monosex culture of *M. nipponense*.

## 1. Introduction

The oriental river prawn (*Macrobrachium nipponense*) is a species of freshwater prawn that belongs to the phylum Arthropods and the subclass Crustacea [1,2]. This species has gained significant popularity in the aquaculture industry in China, Japan, Korea, Vietnam, and Myanmar due to its short breeding cycle, delicious taste, low susceptibility to diseases, adaptability to various environments, and stable market price [3]. However, females of *M. nipponense* undergo periodic and rapid ovarian maturation, particularly during the reproductive season when water temperatures exceed 22 °C [4]. This rapid maturation leads to a prolific multiplication of offspring within the pond, resulting in multigeneration reunion and causing adverse effects such as excessive density, oxygen depletion, and increased feed consumption [5]. This issue may be resolved by the development of reliable monosex cultivation techniques for *M. nipponense* [6]. By selectively producing males or females, population dynamics can be effectively controlled to minimize the negative effects of multigenerational recombination and increase production efficiency [7].

Lipase A (*LIPA*) catalyzes the hydrolysis of cholesterol esters or triglycerides that have been localized within lysosomes following the receptor-mediated endocytosis of low-density lipoprotein particles [8]. The *LIPA* gene is located on chromosome 10 of the human genome, is highly expressed in tissues throughout the body, and contains nine coding exons [9]. Human *LIPA* loss of function is often considered a major cause of recessive lysosomal diseases such as Wolman disease (WOD) and cholesteryl ester storage disease (CESD) [10]. Gene function studies have been conducted on *LIPA* in many other vertebrates such as rhesus monkeys [11], mice [12], rats [13], chickens [14], etc. It has been suggested that the LIP gene family (*LIPA*, *LIPJ*, *LIPK*, *LIPM*, *LIPN*, and *LIPO*) may have been conserved throughout vertebrate evolution and that they have been retained as a separate gene family for more than half a billion years to act in the primary role of acidic lysosomal lipases [15]. In a recent study in mice, the *LIPA* gene was found to be effective in regulating numerous cholesterol-induced problems, including hepatosplenomegaly, elevated serum transaminases, reduced LAL activity, and cholesterol and triglyceride accumulation in organs [16]. Common missense variants in the *LIPA* gene also caused higher levels of triglycerides and hepatic transaminase and lower levels of high-density lipoprotein cholesterol [17].

However, the function of the *LIPA* gene in crustaceans has not yet been studied. In evolutionary history, crustaceans diverged relatively early from other arthropods, and their lineage has distinctive features that differentiate them from vertebrates [18]. This has led to a variety of differences in their physiology and genetic makeup. Therefore, the *LIPA* gene may have different functions in crustaceans. Sex differentiation is the process by which undifferentiated gonads with bidirectional potential are programmed to develop into spermatophores or ovaries and develop secondary sexual traits through a series of events. Eyestalk ablation is commonly used to induce reproductive maturation in crustaceans [19], but it also affects growth and increases energy requirements [20]. This suggests that genes controlling growth are likely to be involved in sex differentiation and gonadal maturation. A previous study on the mechanism of sex determination in *M. nipponense* attempted to identify the “steroid biosynthesis” pathway by comparing the transcriptomes of normal female and unreversed male gonads [21]. A Kyoto Encyclopedia of Genes and Genomes (KEGG) enrichment analysis revealed that the genes in this pathway were significantly enriched in sex determination and gonadal differentiation, and *LIPA* was one of the differentially regulated genes [22].

In this study, the sequence characteristics and phylogenetic relationships of the *Mn-LIPA* gene were analyzed via bioinformatics. The expression pattern of the *Mn-LIPA* gene in different tissues and embryonic developmental stages of *M. nipponense* was investigated via qPCR. The localization of *Mn-LIPA* mRNA in the testes and hepatopancreas was detected via in situ hybridization (ISH) [23]. In addition, the role of the *Mn-LIPA* gene in the regulation of sex differentiation and gonadal development was investigated via RNA interference technology (RNAi). After the knockdown of the *Mn-LIPA* gene by RNAi, the levels of 17β-estradiol (E_2_) [24] and 17α-methyltestosterone (MT) [25] were measured via enzyme-linked immunosorbent assay, and the development of gonads was observed in combination with histological sections, confirming the role of the *Mn-LIPA* gene in gonadal development. Finally, we explored the regulatory role between the *Mn-LIPA* gene and sex-related genes. This study is the first functional exploration of the *Mn-LIPA* gene in crustaceans and provides a solid foundation for understanding the mechanism of sex differentiation and gonadal development in *M. nipponense* and for developing monosex cultivation techniques.

## 2. Results

### 2.1. Molecular Cloning and Structural Analysis of the LIPA Gene

As shown in Figure 1, the full length of the *Mn-LIPA* cDNA sequence was 849 base pairs (bp). The open reading frame was 630 bp and encoded 210 amino acids (GenBank accession number: OR602680.1). Upon analysis of the amino acid sequence of the *Mn-LIPA* gene shown in Figure 2, the estimated protein molecular weight was 23.66 kDa and the isoelectric point was 5.385. The amino acid sequence of the *Mn-LIPA* gene contained three conserved domains, namely, Sp1, Ap2, and CAT, which belong to PLN02872, the only member of the cl28691 superfamily.

BLAST (http://blast.ncbi.nim.nih.gov/Blast.cgi, accessed on 8 October 2023) and phylogenetic analysis of *Mn-LIPA* from *M. nipponense* confirmed that it is related to Insecta and crustaceans. In Figure 3, different colors are used to represent *LIPA* in different families, such as *Pandalidae*, *Palaemonidae*, *Penaeidae*, *Nephropidae*, *Grapsidae*, *Crangonidae*, *Nitidulidae*, *Latreille*, and so on. *Mn-LIPA* forms clusters with various crustaceans, including *Penaeus vannamei*, *Eriocheir sinensis*, *Procambarus clarkia*, and other crustaceans, and is then clustered with insects such as *Aethina tumida*.

### 2.2. Expression of the Mn-LIPA Gene in Different Tissues in Males

*Mn-LIPA* is expressed in the eyestalk, cerebral ganglion, heart, hepatopancreas, testes, and the androgenic gland (Figure 4), but it is almost unexpressed in the gills and muscle. The highest expression level is observed in the hepatopancreas, significantly surpassing other tissues (*p* < 0.05). Following the hepatopancreas, the cerebral ganglion and testes demonstrate moderate expression, while the eyestalk, heart, and the androgenic gland exhibit lower levels of expression.

### 2.3. Expression of the Mn-LIPA Gene during Different Developmental Stages

Figure 5 shows the expression pattern of *Mn-LIPA* at different developmental stages. The expression level of *Mn-LIPA* is highest during the blastula stage, significantly exceeding subsequent stages (*p* < 0.05). Subsequently, it undergoes a rapid decline until the L1 stage. Among the later stages, the highest expression is observed at PL25, followed by PL5, PL10, and PL20.

### 2.4. Localization of the Mn-LIPA Gene in Testes and Hepatopancreas

ISH was used to examine the location of *Mn-LIPA* mRNA in the testes and hepatopancreas (Figure 6). Strong signals were observed in spermatogonium and spermatocyte cells. No signals were observed in sperm cells in Figure 6A. No signals were observed when the negative RNA probe was used. Figure 6B shows the same situation.

### 2.5. Effects of Mn-DCHR RNAi Knockdown on PL10-Stage M. nipponense

#### 2.5.1. Sex Ratio

One hundred PL10-stage *M. nipponense* were injected using *dsMn-LIPA* at a concentration of 8 μg/g. The sex ratio was counted after the third injection (day 15). It is notable that in *M. nipponense*, the glands begin to develop at PL10 and sexual differentiation is complete by PL25, with distinctive secondary sexual characteristics enabling the identification of males and females. As shown in Figure 7, the number of prawns in the experimental and control groups decreased over time. However, the sex ratios of the experimental group for the three stages were 1.00, 0.99, and 1.01, which were not significantly different from the control group (*p* > 0.05).

#### 2.5.2. Steroid Hormone Content

Figure 8 shows the sex hormone content of PL10 *M. nipponense* after five injections. The gonads of *M. nipponense* begin to develop at PL10, and sexual differentiation is completed at PL25 with the emergence of physiological males and females. Therefore, the results are only counted for 15, 22, and 30 days in Figure 8A,B. Figure 8A shows the level of MT in male prawns and Figure 8B shows the level of E_2_ in female prawns. As shown in Figure 8, the MT content in males decreased gradually over time and was consistently lower in the control group (*p* < 0.05). The E_2_ content in females was higher than the control at 15 days and lower than the control at 22 and 30 days (*p* < 0.05).

#### 2.5.3. RNA Interference Efficiency

Figure 9 shows the relative expression of *Mn-LIPA* in the testes of male *M. nipponense* at different stages after the injection of dsRNA. The qPCR analysis showed that the expression of *Mn-LIPA* in the control group remained stable at days 1, 8, and 15 (*p* > 0.05) and displayed a decreasing trend at days 22 and 30. The *Mn-LIPA* expression in the experimental group remained consistently low and was 25.31%, 34.72%, 15.96%, 12.81%, and 10.30% of that in the control group on days 1, 8, 15, 22, and 30, respectively. Throughout the experiment, the expression level of *Mn-LIPA* in the RNAi group was significantly lower than that in the control group (*p* < 0.05).

### 2.6. Effects of Mn-DCHR RNAi Knockdown on Male M. nipponense

#### 2.6.1. Histological Observations of Gonads

As shown in Figure 10A,B, 1 day after the injection, male prawns from both control and *dsMn-LIPA* groups were in the spermatid phase, with primary and secondary spermatocytes in the testes. Fifteen days after the injection, a large number of sperm appeared in the testes of both experimental and control groups (Figure 10C,D). Thirty days after the injection, large numbers of primary spermatocytes, secondary spermatocytes, and sperm were still observed in the control group (Figure 10E). However, as shown in Figure 10F, the testes of the *dsMn-LIPA* group were more mature and almost no spermatocytes were observed.

#### 2.6.2. Steroid Hormone Content

Figure 11 shows the MT content in the testes and hepatopancreas of male *M. nipponense* after five injections. The MT content in the testes and hepatopancreas of male *M. nipponense* in the control group was relatively stable at 374.65 ± 22.82 pb/g and 397.87 ± 24.79 pb/g, respectively. As shown in Figure 11A, in the *dsMn-LIPA* groups, MT in the testes of male *M. nipponense* gradually increased and decreased over time, reaching the highest level of 720.09 pb/g on day 15 (*p* < 0.05). The levels of MT in the hepatopancreas decreased and then increased and were higher on day 1 and day 30 at 687.85 pb/g and 655.61 pb/g (*p* < 0.05), respectively (Figure 11B).

### 2.7. Effects of Steroid Hormones and RNAi on Male Sex-Related Genes in Male M. nipponense

Figure 12 shows the expression of male-related genes in male *M. nipponense* at different stages after 30 days of different treatments (RNAi, MT, and E_2_). After *dsMn-LIPA* injection, insulin-like androgenic gland hormone (*IAG*), sperm gelatinase (*SG*), and mab-3-related transcription factor (*Dmrt11E*) had similar trends of gradually increasing, whereas *Mn-LIPA* expression was significantly downregulated. Sex-related genes, including *Mn-LIPA*, were activated after feeding the prawns with sex hormones, and the expression was consistently higher than in the control group (*p* < 0.05). Negative feedback was shown between *IAG* and *Mn-LIPA* after feeding the prawns with MT, and the expression patterns of *SG* and *Dmrt11E* were similar. The expression of all three genes gradually decreased after feeding the prawns with E_2_, except for *Dmrt11E*, which showed a decrease followed by an increase (*p* < 0.05).

## 3. Discussion

Cholesterol ester lipase (*LIPA*), the only known intracellular lipase active at acidic pH levels, plays an essential role in cholesterol metabolism by hydrolyzing cholesteryl esters and triglycerides within lysosomes and catalyzing the formation of 25-hydroxycholesterol from cholesterol to repress cholesterol biosynthesis [26]. To date, *Mn-LIPA* has only been recognized for its cholesterol-regulating function in vertebrates, but whether it has other functions in crustacean species is not yet known. Crustaceans, like other arthropods, lack the capacity to synthesize the steroid nucleus de novo; therefore, they have to metabolize dietary sterols to form the cholesterol necessary for their growth and reproduction [27]. In this study, we analyzed the function of *Mn-LIPA* in *M. nipponense*, with a particular focus on the role of *Mn-LIPA* in the regulation of sex hormones, sex differentiation, and gonadal development. Our results may be useful in developing a technique to regulate the development of testes in this species.

We successfully cloned the full-length *Mn-LIPA* cDNA from *M. nipponense*. Multiple sequence alignments indicate that *Mn-LIPA* has three nuclear receptor DNA-binding domains (SP1, CAT, and AP2). From the point of view of biological habits, as the fitness of animals living in nutritionally variable environments depends on their ability to modify energy allocation to reproduction, growth, and maintenance in response to cues from the immediate environment, it is likely that the same genes have acted differently in vertebrates and crustaceans over the course of evolution. Phylogenetic analysis showed that crustaceans and insects were clearly delimited and clustered together, indicating that *Mn-LIPA* was differentiated in crustaceans and insects and was more conserved in the same class. To the best of our knowledge, the function of *Mn-LIPA* has not been well defined or analyzed in crustaceans and this is the first qPCR analysis of *Mn-LIPA* expression. We performed qPCR analysis of *Mn-LIPA* in various tissues and at various developmental stages of *M. nipponense*. Tissue-specific analysis revealed that transcripts were detected at much higher levels in the hepatopancreas than in other tissues, followed by the cerebral ganglion and testes, with less expression in the androgenic gland. The hepatopancreas is an important organ in *M. nipponense*. It is responsible for the absorption and metabolization of nutrients, the storage of energy reserves and minerals, the synthesis of lipoproteins for export to other organs, and other functions [28]. This finding suggests a potential role of *Mn-LIPA* in sex differentiation and gonadal development, possibly by controlling steroid synthesis (cholesterol, fatty acids, cholesteryl ester, and triglyceride) and thus influencing the endocrine axis of male reproductive activity, which connects the X-organ-sinus gland (XO-SG), AG, and testes in crustaceans [29].

Previous studies have reported that the sensitive period for sexual differentiation is from PL7 to PL19 [30]. The expression levels of *Mn-LIPA* were highest at the cleavage stage (CS) and PL25 (PL: post-larvae) during the larval and post-larval developmental stages, which are the beginning and end of sex differentiation, respectively, and gradually increased from the blastula stage (BS) to PL20, indicating that it is involved in the process of sexual differentiation in *M. nipponense*. ISH is used to locate a specific DNA or RNA sequence in a tissue or cell using labeled complementary DNA or RNA [31]. In this study, strong *Mn-LIPA* signals were detected in both the testes and hepatopancreas, suggesting that *Mn-LIPA* is involved in steroid synthesis in the hepatopancreas, which in turn promotes sperm maturation and testis development in *M. nipponense*. However, the mechanism of sexual differentiation in decapods is quite complex, and several sex-linked genes and non-coding RNAs have been successively identified, which together form a vast network of interactions that jointly determine sex [32].

RNAi, the most widely used tool in the functional research of crustacean genes, has enhanced our understanding of its target genes in many crustaceans [33,34,35]. In this study, we further confirmed the role of *Mn-LIPA* in sex differentiation and gonadal development of *M. nipponense* using RNAi technology. Oosperm were grown to PL10 and fully sexually differentiated male prawns were subjected to a 30-day RNAi interference experiment. There was no significant change in the sex ratio of PL10 prawns 30 days after a *dsMn-LIPA* injection. However, enzyme-linked immunosorbent assay (ELISA) indicated that levels of MT produced by males in the experimental group were increased whereas levels of E_2_ produced by females were decreased. *Mn-LIPA* expression in the RNAi group was significantly lower than that in the control group on the same day, indicating that dsRNA effectively inhibited *Mn-LIPA* expression. The expression of sex-determining genes is followed by a series of cascading reactions that gradually lead to differentiation in the sex dimension [36]. It has been shown that the sex-determining system is often controlled by multiple genes in lower aquatic animals [37,38]. It is now generally accepted that sexual manipulation in decapods only functions through the manipulation of AG [39], which controls behavior, spermatogenesis, and sexual differentiation through the secretion of IAG. Histological observations showed that *dsMn-LIPA* effectively inhibited the development and maturation of sperm in male *M. nipponense*. Moreover, the experimental group of male prawns had higher levels of MT in the testes and hepatopancreas than the control group. Therefore, we speculate that *Mn-LIPA* is likely to be involved in the formation and physiology of the AG, testes, and hepatopancreas. *Mn-LIPA* is not a master gene for controlling sex and gonadal development, but it can affect sex hormone synthesis. These results suggest that *Mn-LIPA* affects the development of the testes in male *M. nipponense* by regulating hormone levels. To the best of our knowledge, the reproductive function of *Mn-LIPA* has not been focused on in any species before this time.

To study the relationship of *Mn-LIPA* on the regulation of sex-related genes and sex hormones in male *M. nipponense* in more depth, we fed male prawns with 200 mg/kg MT and E_2_ and then used qPCR to characterize the expression of three male-specific genes, *IAG*, *SG*, and *Dmrt11E*. *IAG* is thought to be the active substance responsible for stimulating AG, which encodes and secretes several androgens [40,41]. *SG* is a key player in regulating various reproductive functions, including sperm motility (hyperactivation) [42], acrosome reaction [43], and sperm-egg fusion [44]. *DMRT* proteins share a distinctive zinc-finger DNA-binding motif termed the DM domain, the expression of which is upregulated in association with androgen-induced gonadal masculinization [45,46]. *IAG* and *Dmrt11E* have both been reported to be key genes for sex reversal in *M. rosenbergii* [47,48], and have been studied in *M. nipponense* in relation to male gonadal development [49]. The expression of *IAG*, *SG*, and *Dmrt11E* was increased at different times after the *dsMn-LIPA* injection, which is consistent with the histological observations.

The sex-related genes including *Mn-LIPA* were significantly elevated after feeding prawns with sex hormones, suggesting that *Mn-LIPA* plays a role similar to that of genes such as *IAG*. The expression of *Mn-LIPA* gradually increased after 30 days of MT feeding, in contrast to the expression of *IAG*. Previous studies have shown that exogenous androgens cause changes in gonadal development, but over-reliance on them can lead to a decrease in the self-synthesized secretion of androgens [50,51]. Combining the results of the histological observations and ELISA, we speculate that *Mn-LIPA* does not directly control the production of sex hormones but rather utilizes the property of hydrolyzing triglycerides and cholesterol to provide energy while influencing the synthesis and secretion of self-sex hormones [26,52]. The expression of *IAG*, *SG*, and *Mn-LIPA* in the experimental group gradually decreased with the feeding of E_2_ but was always higher than the control group. This suggests that *Mn-LIPA* is activated by sex hormones, similar to other sex genes. This may be one reason why feeding prawns with sex hormones can cause weight gain [53,54], and the expression of genes in the E_2_ group also supports this conjecture [55,56]. However, our study did not expose the reasons behind this, such as how *Mn-LIPA* affects the physiological activities of organs like AG, how *Mn-LIPA* regulates MT and E_2_ secretion, whether *Mn-LIPA* has similar functions in vertebrates, etc.

## 4. Materials and Methods

### 4.1. Experimental Prawns

Thirty healthy egg-holding female *M. nipponense* (body weight = 4.42 g ± 0.55 g) were obtained from Taihu Lake (Wuxi, China; 120°13′44′′ E, 31°28′22′′ N) and cultured until they hatched young prawns. Larvae were fed with Artemia until their body weight reached PL10 (0.0092 ± 0.0017 g) and PL30 (PL: post-larvae) male *M. nipponense* (1.4351 ± 0.1621 g), and then they were assigned randomly to the control and experimental groups. As shown in Figure 13, an injection was performed every 7 days; after each injection, samples of testes were collected for tissue observation, steroid hormone levels and sex ratios were measured, and qRT-PCR was performed.

### 4.2. Nucleotide Sequencing and Bioinformatics Analysis

The total RNA of *M. nipponense* was extracted using RNAiso Plus Reagent (TaKaRa, Dalian, China). The full-length *Mn-LIPA* cDNA sequence was obtained from the testis transcriptome library of female *M. nipponense* (accession number: OR602680) under Bioproject PRJNA830321. Three pairs of primers (1, 2, 3) were used to validate the ORF. The *Mn-LIPA* gene sequence was analyzed using GenBank BLASTX and BLASTN programs (http://www.ncbi.nlm.nih.gov/BLAST/, accessed on 12 October 2023). The open frame analysis of the full-length cDNA sequence of the obtained *Mn-LIPA* gene was performed using the online program ORF Finder (http://www.ncbi.nlm.nih.gov/gorf/gorf.html, accessed on 12 October 2023). The sequences were subjected to BLAST alignment to test for similarity and consistency. ExPASy ProtParam (http://web.expasy.org/compute_pi/, accessed on 12 October 2023) was used to predict the isoelectric points and mean relative molecular mass of the *Mn-LIPA* amino acid sequence. The phylogenetic tree analysis was performed using MEGA 5.0, and the calculation was repeated 1000 times using the bootstrap method. Multiple sequence alignment of *Mn-LIPA* amino acids was performed using DNAMAN 6.0 software.

### 4.3. Tissue Expression Analysis by Quantitative Real-Time PCR

The Bio-Rad iCycler iQ5 Real-Time PCR system (Bio-Rad, Hercules, CA, USA) was used to detect the expression of the *Mn-LIPA* gene in the above samples [57]. The specific primers are listed in Table 1. EIF was selected as the reference gene for qPCR [58]. The qPCR expression level of *Mn-LIPA* was calculated via the 2^−ΔΔCT^ method [59].

### 4.4. In Situ Hybridization (ISH) and Histological Observations

The testes of mature male *M. nipponense* were collected and fixed in 4% paraformaldehyde in PBS (pH 7.4) at 4 °C overnight. Following the method of Satoet al. [60], we used the Zytofast PLUS CISH implementation kit and embedded the samples in paraffin. Primers were designed using Primer 5.0 software based on the *Mn-LIPA* cDNA sequence. The sequence 5′-CCTCATCATTTCCAAATAATCTCAGGACAAACTG-3′ was used as the ISH probe. The slides were examined under a light microscope.

Hematoxylin and eosin (HE) staining was used in order to study the histological changes in the testes and ovaries. On days 1, 15, and 30 after injection, testes samples from the control and *Mn-LIPA* groups were mounted on slides, which were stained via hematoxylin and eosin staining followed by cleaning with different concentrations of alcohol [61]. Observations took place using a stereo microscope (SZX16; Olympus Corporation, Tokyo, Japan). Comparative labeling was performed with various cell types based on cell morphology [21].

### 4.5. Sex Ratio Statistics

The prawns in the control and experimental groups were randomly assigned with more than 90 individuals in total to measure the sex ratio and feminization rate. Each group had at least three replicates.

### 4.6. RNA Interference of LIPA

The online software Snap Dragon (http://www.flyrnai.org/cgi-bin/RNAi_find_primers.pl, accessed on 20 October 2023) was used to design RNAi primers. The T7 promoter sequence was added to the 5′ end of the RNAi primer sequence. T7: TAATACGACTCACTATAGGG. Then, the *Mn-LIPA* fragment was amplified using RNAi primers. After detection, double-stranded RNA was synthesized using the Transcript AidTMT7 high yield transcription kit (Fermentas, Waltham, MA, USA). The dsRNA was verified using standard agarose gel electrophoresis and the concentration of dsRNA was measured using a BioPhotometer (Eppendorf, Hamburg, Germany). Finally, it was stored at −80 °C.

### 4.7. Enzyme Linked Immunosorbent Assay (ELISA)

The content of MT in the testes of male prawns and E_2_ in the ovaries of female prawns were detected by ELISA after RNAi. Tissues from experimental and control groups were collected on days 1, 8, 15, 22, and 30 post-injection, snap-frozen in liquid nitrogen, and stored at −80 °C until MT and E_2_ content were determined (n = 9). The sex hormone content in prawns was determined using double-antibody one-step sandwich ELISA following the instructions of the Shrimp EH ELISA Kit (Lot number: 20230724-YJ923014 and 20230724-YJ950014; Mlbio Shanghai, Shanghai, China).

### 4.8. Dietary Preparation

The diets used in this study came from the commercial prawn diet guidelines produced by the Freshwater Fisheries Research Center, Chinese Academy of Fisheries Sciences, China (Wuxi, China). The commercial diet is mainly composed of crude protein, fish meal, shrimp meal, squid meal, starch, soybean meal, ash, canola meal, soybean protein concentrate, crude lipid, etc. [62]. The MT (CAS number: 58-18-4, purity: 97.65%) and E_2_ (CAS number: 50-28-2, purity: 95.88%) were purchased from Beijing Solarbio Technology Co, Ltd. (Beijing, China). The method of dissolving hormones into the diets is described below [63]. Hormones were dissolved in 95% ethanol to prepare a stock solution at a concentration of 20 mg/mL, and then the solution was evenly sprayed on the feed (1 mL ethanol per 10 g diet) and stirred with a glass stick for at least three minutes. After that, it was placed under a ventilated laboratory hood and left in shade for 15 min. The treated feed was poured into 15 mL test tubes and placed in a refrigerator at 0 °C to evaporate the remaining alcohol naturally.

### 4.9. Data Analysis

The statistical analyses were all conducted using IBM SPSS Statistics for Windows, version 23.0. (IBM Corporation, Armonk, NY, USA). The significant differences between groups were determined via one-way ANOVA, followed by the least significant difference test and Tukey’s test [64]. Quantitative data are expressed as mean ± standard deviation (SD). Probability (*p*) values < 0.05 were considered statistically significant.

## 5. Conclusions

This study describes for the first time the role of *Mn-LIPA* in sex hormone regulation and gonadal development in *M. nipponense*. We found that *Mn-LIPA* is highly expressed in the hepatopancreas, cerebral ganglion, and testes, indicating its involvement in sex differentiation. ISH revealed *Mn-LIPA* signaling in the spermatheca and hepatopancreas, suggesting its role in steroid synthesis and sperm maturation. RNAi experiments confirmed the impact of *Mn-LIPA* on sperm development. Furthermore, this study demonstrated the relationship between *Mn-LIPA* and sex-related genes and hormones. Overall, these findings enhance our understanding of the function of *Mn-LIPA* in crustaceans and its effects on sex differentiation and gonadal development. This study provides an important theoretical basis for the realization of a monosex culture of *M. nipponense*.

## Figures and Tables

**Figure 1 ijms-25-01399-f001:**
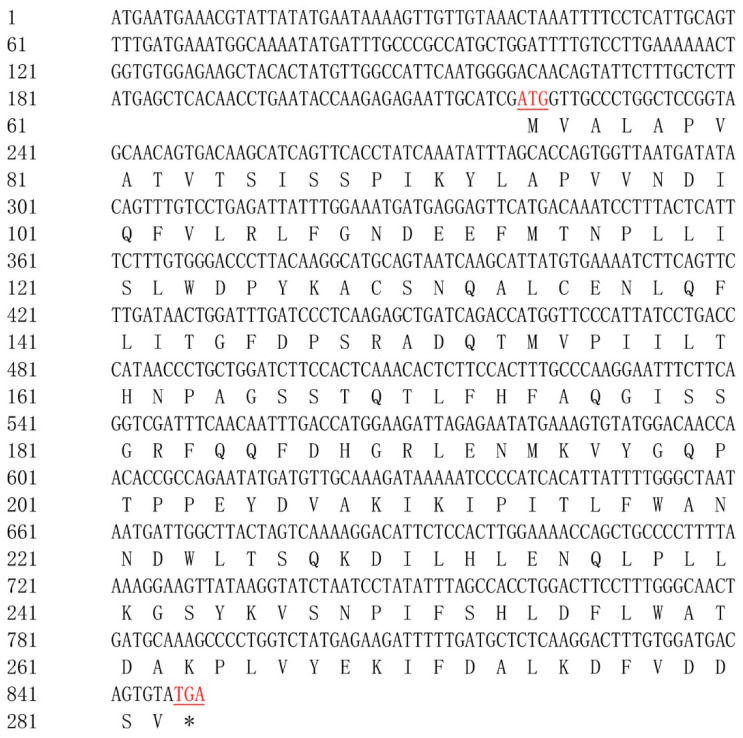
The nucleotide and deduced amino acid sequences of *Mn-LIPA*. The numbers on the left of the sequence indicate the positions of nucleotides and amino acids. The start codon ATG and the stop codon TGA are indicated in red and underlined, and the TGA stop codon in the amino acid sequence is represented by an asterisk (*).

**Figure 2 ijms-25-01399-f002:**
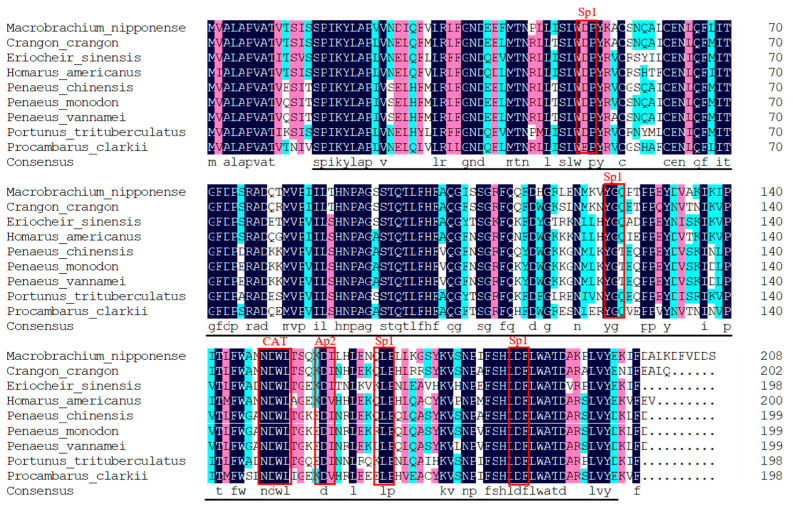
Amino acid sequences from other species aligned with the sequence encoded by the *Mn-LIPA* gene. The black line indicates the PLN02872 superfamily. The red boxes mark the three conserved domain sites. GenBank accession number: *Macrobrachium nipponense* (OR602680.1), *Crangon crangon* (MH055770.1), *Eriocheir sinensis* (OU618541.1), *Homarus americanus* (XM_042384002.1), *Penaeus chinensis* (XM_047625243.1), *Penaeus monodon* (XM_037925897.1), *Penaeus vannamei* (XM_027357916.1), *Portunus trituberculatus* (XM_045258007.1), and *Procambarus clarkia* (XM_045760019.1).

**Figure 3 ijms-25-01399-f003:**
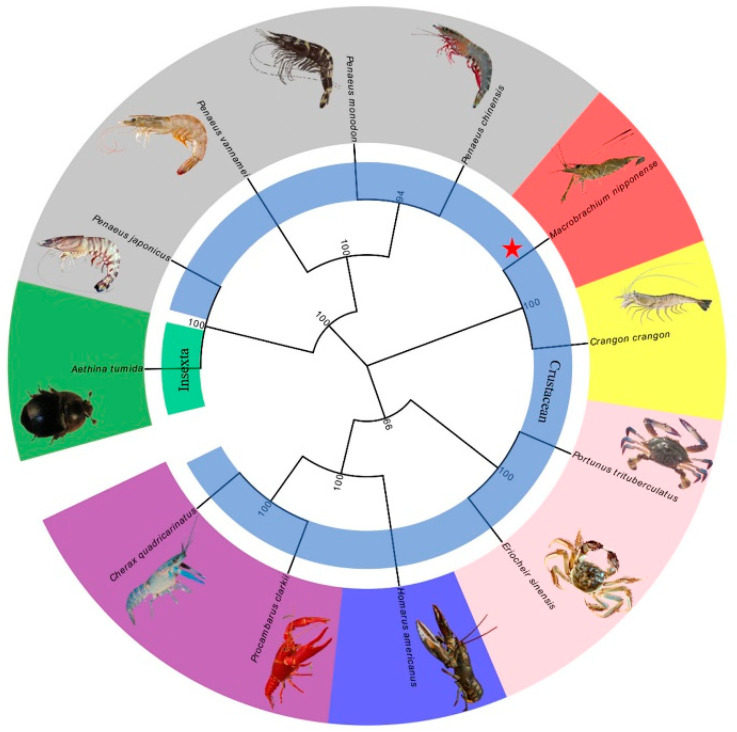
Phylogenetic tree of *Mn-LIPA* sequences. The diagram was generated via the neighbor-joining method using the MEGA 7.0 program, with 1000 bootstrapping replications. The red star indicates the location of the *M. nipponense*. The numbers indicate the reliability of branches.

**Figure 4 ijms-25-01399-f004:**
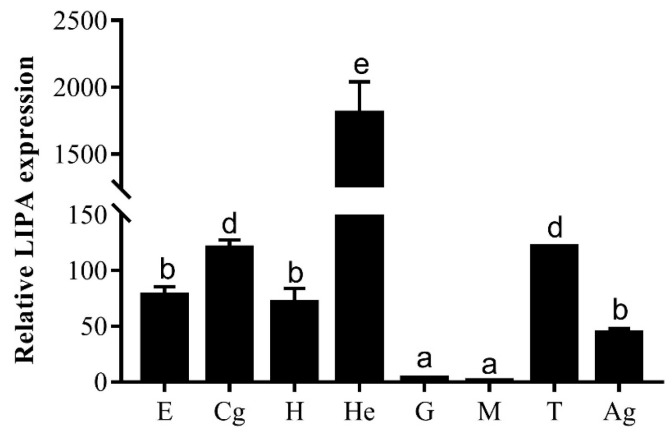
Expression pattern of the *Mn-LIPA* gene in different tissues. E: eyestalk, Cg: cerebral ganglion, H: heart, He: hepatopancreas, G: gill, M: muscle, T: testes, Ag: androgenic gland. Data are presented as the mean ± SD (n = 6). Different letters indicate significant differences (*p* < 0.05) which were analyzed via one-way ANOVA followed by least significant difference and Tukey’s test.

**Figure 5 ijms-25-01399-f005:**
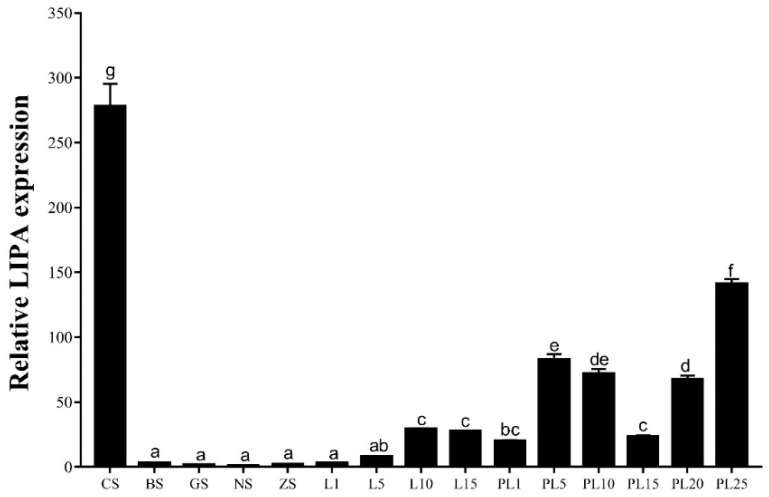
The expression levels of *Mn-LIPA* during embryonic development stages. CS: cleavage stage; BS: blastula stage; GS: gastrula stage; NS: nauplius stage; ZS: zoea stage; L1: the first day after hatching; PL1: the first day post-larvae, and so on. Data are shown as mean ± SD (n = 6). Different letters indicate significant differences (*p* < 0.05) which were analyzed by one-way ANOVA followed by least significant difference and Tukey’s test.

**Figure 6 ijms-25-01399-f006:**
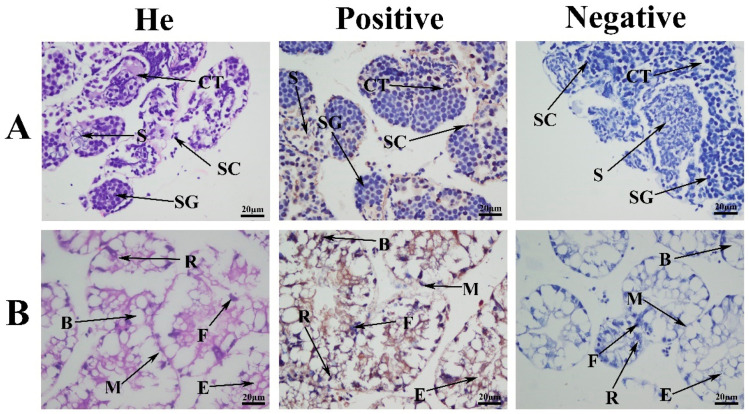
In situ hybridization analysis of *M. nipponense*. (**A**) Testes; (**B**) hepatopancreas. SG: spermatogonium; SC: spermatocyte; S: sperm; CT: collecting tissues, E: embryonic cell, R: resorption cells; F: fiber cells; B: blister cells, M: midget cells. Scale bars = 20 μm.

**Figure 7 ijms-25-01399-f007:**
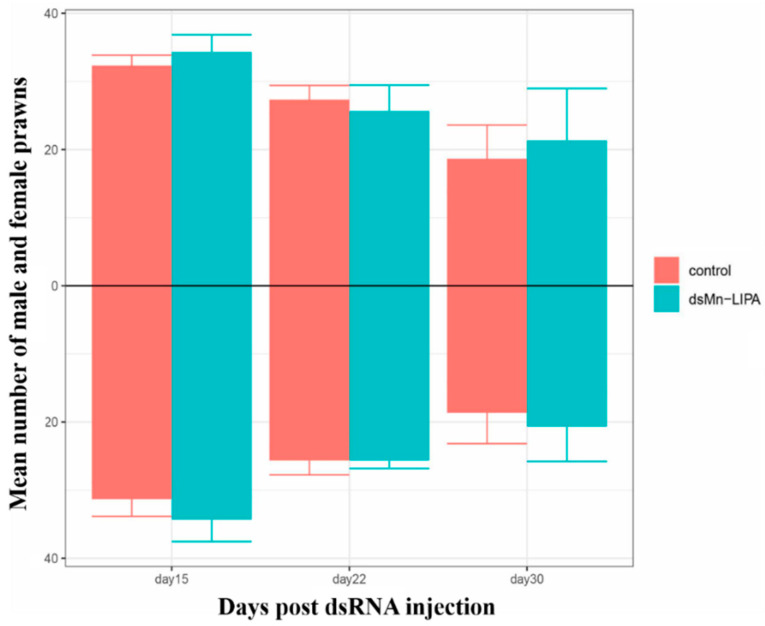
The number of male and female prawns on days 15, 22, and 30 after the injection of PL2 *M. nipponense*. The number of males is above the black line and the number of females is below. Data are presented as the mean ± SD (n > 20).

**Figure 8 ijms-25-01399-f008:**
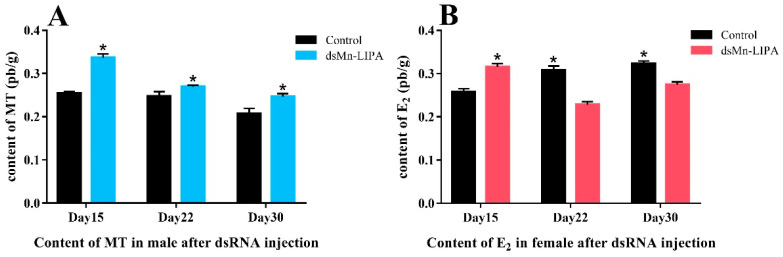
Sex hormone levels in males and females after PL10 *M. nipponense* received five injections of dsRNA. (**A**) Levels of 17α-methyltestosterone in males; (**B**) Levels of 17β-estradiol in females. Data are presented as the mean ± SD (n = 6); * indicates statistical significance at *p* < 0.05, which were analyzed by one-way ANOVA followed by least significant difference and Tukey’s test.

**Figure 9 ijms-25-01399-f009:**
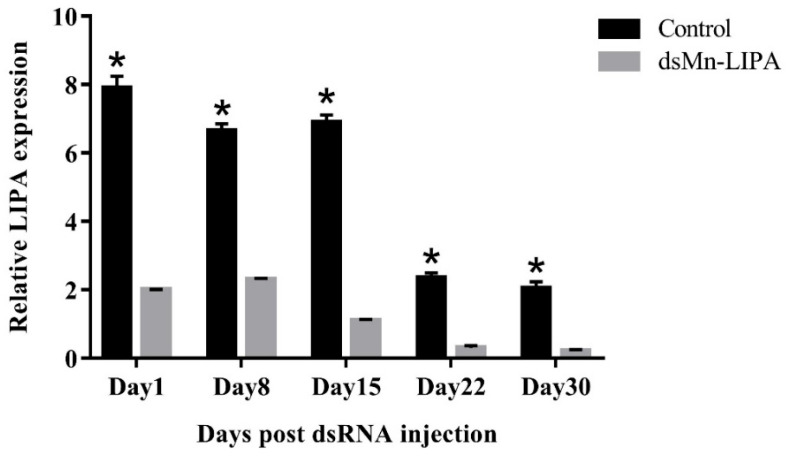
Expression levels of *Mn-LIPA* in the testes after injection with *dsMn-LIPA*. Data are shown as mean ± SD (n = 6); * indicates statistical significance at *p* < 0.05, which were analyzed by one-way ANOVA followed by least significant difference and Tukey’s test.

**Figure 10 ijms-25-01399-f010:**
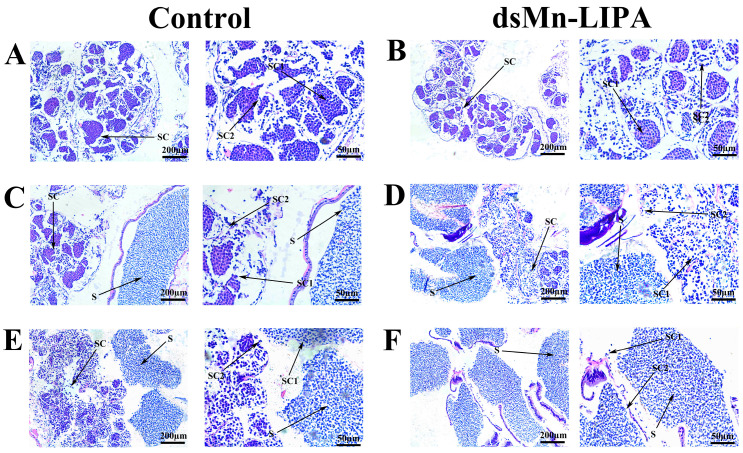
Histological sections of testes from control and *dsMn-LIPA* groups after five injections. (**A**,**B**) Histological sections of testes on day 1; (**C**,**D**) Histological sections of testes on day 15; (**E**,**F**). Histological sections of testes on day 30. SC: Spermatocyte; SC1: Primary spermatocyte; SC2: Second spermatocyte; S: Sperm. Scale bars: 200 µm and 50 µm.

**Figure 11 ijms-25-01399-f011:**
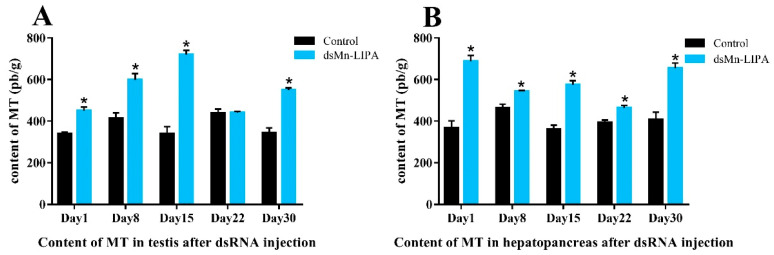
Levels of 17α-methyltestosterone in the testes and hepatopancreas of male *M. nipponense* after five injections of dsMn-LIPA. (**A**) Level of 17α-methyltestosterone in the testes; (**B**) Content of 17α-methyltestosterone in hepatopancreas. Data are presented as the mean ± SD (n = 6); * indicates statistical significance at *p* < 0.05, which were analyzed by one-way ANOVA followed by least significant difference and Tukey’s test.

**Figure 12 ijms-25-01399-f012:**
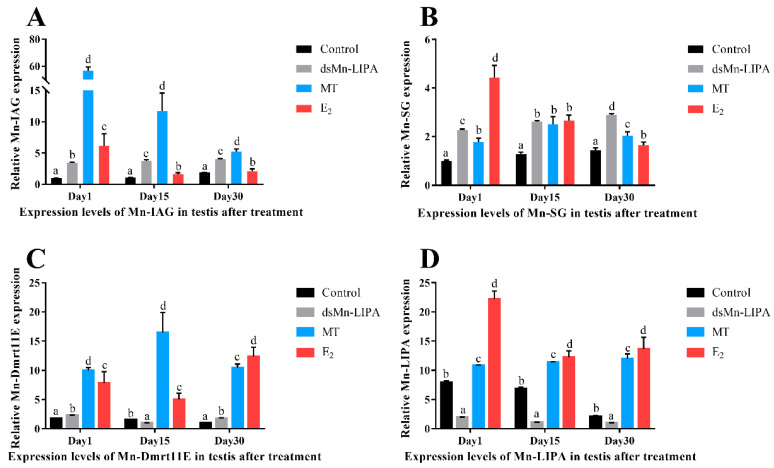
Expression levels of sex-related genes in testes after treatment with *dsMn-LIPA*, 17α-methyltestosterone, and 17β-estradiol. (**A**) *Mn-IAG*; (**B**) *Mn-SG*; (**C**) *Mn-Dmrt11E*; (**D**) *Mn-LIPA*. Data are shown as the mean ± SD (n = 6); different letters indicate significant differences (*p* < 0.05) which were analyzed by one-way ANOVA followed by least significant difference and Tukey’s test.

**Figure 13 ijms-25-01399-f013:**
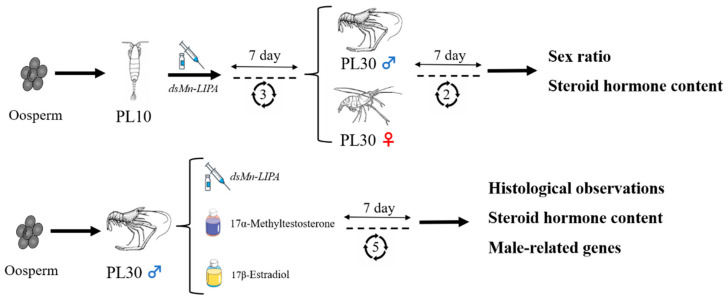
Experimental design.

**Table 1 ijms-25-01399-t001:** Primers for cDNA clone, qPCR analysis, and RNAi that were involved in this study.

Primer Name	Sequence (5′→3′)	Usage
*LIPA*-F1	ACTCATTTCTTTGTGGGACCCTT	ORF
*LIPA*-R1	GAAATTCCTTGGGCAAAGTGGAA	ORF
*LIPA*-F2	CGTAGCAACAGTTACAAGCATCA	ORF
*LIPA*-R2	AAGGGTCCCACAAAGAAATGAGT	ORF
*LIPA*-F3	ACTCAAACACTCTTCCACTTTGC	ORF
*LIPA*-R3	ATAGACCAGAGGCTTTGCATCAG	ORF
*Mn-LIPA*-F	ACCGCCAGAATATGATGTTGCTA	qPCR
*Mn-LIPA*-R	AAAGGAAGTCCAGGTGGCTAAAT	qPCR
*Mn-IAG*-F	CGCCTCCGTCTGCCTGAGATAC	qPCR
*Mn-IAG*-R	CCTCCTCCTCCACCTTCAATGC	qPCR
*Mn-SG*-F	ACCCTAGCCCCAGTACGTGTT	qPCR
*Mn-SG*-R	AGAGGTGGTGAAGCTGTCTCTCA	qPCR
*Mn-Dmrt11E*-F	ACGACCTTAGTAGGATGGACAGT	qPCR
*Mn-Dmrt11E*-R	GAGTGGAGGCAATAGAATGGGTA	qPCR
*Mn-EIF*-F	CATGGATGTACCTGTGGTGAAAC	qPCR
*Mn-EIF*-R	CTGTCAGCAGAAGGTCCTCATTA	qPCR
*dsMn-LIPA*-F	TAATACGACTCACTATAGGGACTGACCCATAACCCTGCTG	dsRNA
*dsMn-LIPA*-R	TAATACGACTCACTATAGGGTCAGTTGCCCAAAGGAAGTC	dsRNA

Note: GenBank accession number: *Mn-LIPA* (OR602680.1), *Mn-IAG* (JX962354), *Mn-SG* (EF647641), *Mn-Dmrt11E* (MH636338), *Mn-EIF* (MH540106).

## Data Availability

The data presented in this study are available on request from the corresponding author for scientific purposes.

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
