# Peer review of "Role of Mn-LIPA in Sex Hormone Regulation and Gonadal Development in the Oriental River Prawn, Macrobrachium nipponense"

_ijms, 2024, doi:10.3390/ijms25031399_

Round 1

Reviewer 1 Report

Comments and Suggestions for Authors

Comments to the Author

Over the manuscript, the English quality is inferior. Many spelling, grammatical, punctuation, and typographical errors are found throughout the manuscript. Please improve your academic English.

Abstract

·      The abstract must contain the overall purpose of the study and the research problem(s) you investigated, major findings, and conclusions. The abstract should state briefly the methodology of the research, the principal results, and major conclusions. Also, non-standard or uncommon abbreviations should be avoided, but if essential they must be defined at their first mention in the abstract itself. Please rewrite the abstract.

The introduction

·      I believe the introduction is not coherent enough. Please explain a few general statements about the subject to provide a background to your essay and to attract the reader's attention. The introduction should be rewritten about the design of the study.

Results

1- Figure 6, the resolution, and staining are very bad, and the figure contains many artifacts.

2-In Figures 9 & 12, what are the methods used to evaluate relative expression fold change or what??? You should clarify this in the figure.

·       3- Results must be significantly improved. There is no flow in the writing, and the authors in some sections mention the p-value level and other sections do not.

Discussion

1- Authors must be discussing your results; this type of discussion is not reasonable and not solid for a real discussion. The discussion part is too vague. Compare your results with previous similar findings with relevant justifications.

·       2-The authors should mention the limitations of their findings and their perspectives.

·       3-Please improve your conclusion it's poorly written, and it doesn’t explain the results of this study, please improve.

Material and methods

·      ARRIVE guideline; In order to maximize the quality and reliability of this research, I suggest this animal research should comply with all 21 items of the ARRIVE guideline. Please follow the guidelines from the link below: https://arriveguidelines.org/arrive-guidelines

·      All accession numbers of primers should be mentioned.

·      I am very confused about your methods what type of samples did you use to evaluate MT and E2 in prawn tissues and how did you handle them?

·       Statistic – please, provide information about tests by which Gauss distribution has been evaluated.

Comments on the Quality of English Language

Over the manuscript, the English quality is inferior. Many spelling, grammatical, punctuation, and typographical errors are found throughout the manuscript. Please improve your academic English.

Author Response

We thank the reviewers for their time and effort spent to critically review our manuscript. Based on these comments and suggestions, we have made careful modifications on the manuscript. All the changes made to the text are in purple color. Below, we attached a point by point response to all questions and concerns.

Reviewer 2 Report

Comments and Suggestions for Authors

I propose to publish the article after making all the corrections I request.

After reading the work I am not sure that I am suitable to be a reviewer in this work.

My specialty is in fish (vertebrates and not Invertebrates) and the genes that control

reproduction. If another cancer researcher thinks like me, I recommend accepting the job after correcting my comments.

I trust you to check the corrections, please don’t send me again this article. It was difficult for me to judge because there are no numbers for the rows. Hope my comments will be will be understood.

Abstract correct the Abstract

The first time a species name or any other abbreviation appears, the entire name must

be written - Macrobrachium nipponense – (M. nipponense),

qPCR, Mn-LIPA …….

Role in steroid synthesis. Please explain these are steroids and not a general word.

Sex-related genes and hormones was explored. Please be more precise, and not

general words that include a large number of genes and hormones.

Introduction

KEGG - See previous comment regarding abbreviations and correct in all work I will

not repeat this comment.

No citations were given in the last paragraph of the introduction, although I know

there is information. Please emphasize what is known about crabs before this work.

Please add more information on sex determination and differentiation center of this

work. The description of the hormones that control the processes is also important.

Please make it clear to the reader what the new findings are that will not be shown

in the previous work of those authors.

Identification of Important Genes Involved in the Sex-Differentiation Mechanism of

Oriental River Prawn, Macrobrachium nipponense, During the Gonad Differentiation

and Development Period

www.frontiersin.orgShubo Jin www.frontiersin.orgWenyi Zhang

www.frontiersin.orgYiwei Xiong www.frontiersin.orgSufei Jiang

www.frontiersin.orgHui Qiao www.frontiersin.orgYongsheng Gong

www.frontiersin.orgYan Wu www.frontiersin.orgHongtuo Fu*

Results

Fig. 4, 5 ….12. Different letters indicate significant differences (p < 0.05). Add here

the name of the statistical analysis in which the results were compared.

Discussion

I recommend improving the discussion.

To date, the actual functions of Mn-LIPA have not been reported for any crustacean species.

This is true? Check again.

The first three chapters are a repetition or explanation of the results. In the

discussion, the results should not be repeated, the innovations should be explained

in comparison to what was known before and the contribution of this work should

be emphasized.

The subject of this study is -Gene in Sex Hormone Regulation and Gonadal Development in

the Oriental River Prawn, Macrobrachium nipponense - I recommend trying to present a

quality model from what is found in this work and what is known in other crustacean

species that describes the effect of the hormones encoded in the genes on the

development of the genitals in males and females.

Author Response

We thank the reviewers for their time and effort spent to critically review our manuscript. Based on these comments and suggestions, we have made careful modifications on the manuscript. All the changes made to the text are in bule color. Below, we attached a point by point response to all questions and concerns.

Reviewer 3 Report

Comments and Suggestions for Authors

The manuscript (MS) was well designed for the aim and had the first attempt for the first time on the topic. The MS studied all detailed gene, tissue, in the experiment considering LIPA addition as compared to the background null experiment to test its effect on the gonadal development with sex hormone regulation of the commercially very important freshwater prawn.  

Nevertheless, the text of MS needs fluent sentences more than the current version, 

In the text, there are syntax errors in citations given in the text. 

I recommended that the MS could be accepted after minor revision. 

Comments on the Quality of English Language

The text of MS needs fluent sentences more than the current version, 

There are some nouns formed with 4-5 successive words, which makes them hard to read. The author can rephrase such sentences. 

Author Response

We thank the reviewers for their time and effort spent to critically review our manuscript. Based on these comments and suggestions, we have made careful modifications on the manuscript. The manuscript has been sent to a native English speaker for edited and corrected all grammatical, spelling, and other errors.

Round 2

Reviewer 1 Report

Comments and Suggestions for Authors

Dear authors,

Please apply all responses to comments using track change so I can easily found. It is difficult for me to recognize the corrections you made.

Comments on the Quality of English Language

 Moderate editing of English language required

Author Response

(The authors gave the same response as above.)

Round 3

Reviewer 1 Report

Comments and Suggestions for Authors

Thank you for applying all recommended comments 

Comments on the Quality of English Language

Minor editing of English language required